# Galileo-Based Doppler Shifts and Time Difference Carrier Phase: A Static Case Demonstration

**DOI:** 10.3390/s23156828

**Published:** 2023-07-31

**Authors:** Ciro Gioia, Antonio Angrisano, Salvatore Gaglione

**Affiliations:** 1Independent Researcher, 21020 Brebbia, Italy; 2Department of Engineering, Messina University, 98121 Messina, Italy; antonio.angrisano@unime.it; 3Department of Science and Technology, University of Naples Parthenope, 80143 Napoli, Italy; salvatore.gaglione@uniparthenope.it

**Keywords:** Galileo, GNSS-based velocity, Doppler, TDCP

## Abstract

The European Commission is designing and implementing new regulations for vehicle navigation in different sectors. Commission Delegated Regulation 2017/79 defines the compatibility and performance of the 112-based eCall in-vehicle systems. The regulation has a large impact on road transportation because it requires that all cars and light duty vehicles must be equipped with eCall devices. For heavy duty vehicles, a set of new regulations has been developed, starting from EU Regulation No 165/2014, in which the concept of smart tachographs was introduced to enforce the EU legislation on professional drivers’ driving and resting times. In addition, intelligent speed assistance (ISA) devices increase the safety of road users. These new devices fully exploit the Global Navigation Satellite System (GNSS) to compute position velocity and time (PVT) information. In all these systems, the velocity of the vehicle plays a fundamental role; hence, a reliable and accurate velocity estimate is of utmost importance. In this work, two methods for velocity estimation using Galileo are presented and compared. The first exploits Doppler shift measurements, while the second uses time difference carrier phase (TDCP) measurements. The Doppler-based technique for velocity estimation is widely adopted in current devices, while the TDCP technique is emerging due to its promising high accuracy. The two methods are compared considering all the Galileo signals including E1, E5a, E5b, E5 Alt BOC and E6. The methods are compared in terms of velocity errors for both horizontal and vertical components using real static data. From the tests performed, it emerged that the TDCP has increased performance with respect to the Doppler-based solution. Among the Doppler-based solutions, the most accurate solution is the one obtained with the E5 Alt BOC signal.

## 1. Introduction

People and goods are transported by millions of vehicles (cars, buses, trucks, vans) on extensive national and international road transport infrastructure. Road transport strongly contributes to economies in terms of Gross Domestic Product (GDP) and employs almost 5 million people in the EU alone [1]. Therefore, safer and more secure vehicle navigation is essential to improve the safety of people travelling or working on roads. The improvement in technology employed in vehicles could contribute to the safety of mobility if the systems applied are accurate and robust. In this respect, the European Commission is designing and implementing new regulations for vehicles navigation.

Commission Delegated Regulation 2017/79 stipulates that from the 31 March 2018 all new models of passenger cars and light duty vehicle types had to be equipped with eCall in-vehicle systems [2]. In the event of an accident, in-vehicle sensors will automatically trigger an emergency call. An audio connection is made with the European emergency number 112 and routed to the Public-Safety Answering Point (PSAP). At the same time, an emergency message is sent, providing information including the time, location and driving direction. The eCall device technology is based on the Global Navigation Satellite System (GNSS) for vehicle navigation parameters estimation and on mobile communication. The type of device needs to be approved by authorized technical centers; among the tests to be passed by the eCall devices, there are specific tests in dynamic and static conditions. Results of a testing campaign performed by the Joint Research Centre (JRC) were presented in [3]. A more proactive system based on similar technology is the smart tachograph, which has been mandatory for European professional drivers since 2016 thanks to the entry into force of EU Regulation 165/2014. On 18 March 2016, the European Commission adopted legislation defining new technical specifications for smart tachographs. The Smart Tachograph version 2 will improve robustness against GNSS jamming and spoofing attacks by using Open Service Navigation Message Authentication (OS-NMA) from Galileo. The Smart Tachograph version 2 will be mandatory for newly registered vehicles as of 21 August 2023 and for all vehicles involved in international transport as of 21 August 2025. In this case, the smart tachograph is responsible for recording various aspects of a driver’s work, including driving times, rest periods, and periods of other work and availability. The measure of all these parameters plays a key role in preventing driver fatigue, improving road safety [4]. The device should be able to measure the speed with a tolerance of ±1 km/h (at constant speed) [5].

Among all the risk factors related to road safety, speeding is almost universally recognized as the most significant [6]. Speed is a major contributor to the likelihood of traffic accidents as well as the severity of the injuries they cause. Speed can be misleading because many factors, such as the type of vehicle, the time of day, the weather, the design and condition of the road, all influence how it is perceived as a risk factor. Speeding makes it more likely that a driver will lose control of the car because it gives them less time to see impending dangers. Additionally, it leads other road users to mistake the intentions of the speeding driver. Therefore, it is important to accurately estimate this parameter in post-crash analysis performed for insurance or investigation purposes.

Different methods can be used for velocity estimation. Automated driving systems (ADS) vehicles are equipped with multi-modal sensors including LiDAR, cameras, and GNSS. In [7], an ADS data acquisition and analytics platform for vehicle trajectory extraction, reconstruction, and evaluation based on connected automated vehicle (CAV) cooperative perception is presented. HYDRO-3D [8] combines object detection features from a state-of-the-art object detection algorithm with historical information from the object tracking algorithm to infer objects. A novel algorithm referred as YOLOv5-tassel is proposed in [9] with the aim to detect tassels in Unmanned Aerial Vehicle (UAV)-based Red Green Blue (RGB) imagery. For the integration of LiDAR and camera, synchronization is fundamental. Usually the time stamp is obtained from the Pulse Per Second (PPS) generated by the GNSS receiver; in order to obtain a reliable time, the GNSS receiver should use barriers against interference [10], and the relevance of the integrity algorithm for timing the GNSS receiver is shown in [11].

The velocity estimation performed by all the smart devices, which is a mandatory requirement of European rules, is based on GNSS technology. GNSS is relatively important for vehicle safety as it can provide the vehicle position, velocity, and attitude with the fusion of IMU and other sensor signals, as demonstrated in [12,13,14]. However, GNSS receivers in transportation operate mainly in single-point position mode, and different approaches are possible for velocity estimation. The simplest approach consists in differencing two consecutive positions (i.e., approximating the derivative of user position): this method is the easiest to implement, but because of its reliance on pseudorange-based position, it has an accuracy level of one meter per second [15]. Alternatively, receiver velocity can be estimated by using Doppler measurements related to user–satellite motion. Specifically, as a result of the user and satellite’s relative motion, the received signal experiences Doppler frequency shifts, which allows accurate determination of velocity in the order of a few centimeters per second [15,16,17].

Such accuracy can be insufficient for several applications, including highly dynamic mobile navigation. Processing the differences between successive carrier phase measurements (TDCP, or time difference carrier phase) can lead to performance improvements and enable velocity accuracies that are one order of magnitude better than Doppler measurements [17,18,19].

The TDCP observables have a direct relationship with the position increment (delta position), making them useful for calculating the average velocity between two given epochs. The effects of various common errors between the measurements are eliminated or significantly diminished when two consecutive carrier phases are differentiated, enabling a very precise velocity estimation.

A drawback of TDCP is that it can exhibit lower solution availability compared with the Doppler-based technique, especially in harsh environments like urban canyons. In fact, TDCP velocity computation requires at least four continuous carrier phases (single constellation case) for two consecutive epochs, whereas Doppler-based velocity only requires the availability of four Doppler measurements at the considered epoch. A further drawback of TDCP is its sensitivity to cycle slips which can occur and results in velocity errors of several centimeters per second. To solve this issue, a technique for the detection and correction of cycle slips should be used prior to differentiating two successive carrier phases. Additionally, the traditional broadcast ephemeris selection criterion is inappropriate for TDCP. Typically, the set of broadcast ephemeris nearest to the measurement epoch is chosen to compute the position with pseudoranges or velocity with Doppler measurements. In this way, it is possible that two distinct ephemeris sets are used in two specific consecutive epochs. Because code positioning and Doppler-based velocity estimation are “snapshot” techniques, they are not affected by this event. The TDCP technique, on the other hand, necessitates two consecutive measurements, and the use of different broadcast ephemerides may result in a discontinuity in the TDCP measurements, which may then affect velocity estimation [20].

In this study, the performance of Doppler shift and TDCP techniques is investigated using all frequency measurements of the Galileo system. The aim of the paper is to compare these two different techniques, one snapshot and one time differencing, investigating the opportunity provided by the different measurements of the Galileo system [21,22]. Galileo satellites transmit signals on E1, E5 and E6 frequencies. The E5 signal is further sub-divided into signals denoted as E5a, E5b and E5 Alt BOC. For all these frequencies, Doppler shifts and carrier phases were processed by the developed algorithm. The Doppler-based technique is widely used, while the TDCP technique is rapidly gaining popularity. The authors do not know of any other research which analyses the aforementioned techniques with all the available Galileo signals. The algorithm has a structure similar to PANG-NAV toolbox that already computes the Doppler-based velocity [23].

The performance of the methods is assessed using real data collected in static conditions. A long data collection period of about 24 h using a professional receiver was used for testing. Although, static conditions reduce the presence of issues such as cycle slips and multipath propagation, they have the advantage of an accurate reference solution, which allows proper error characterization for the methods presented.

The paper is structured as follows. In Section 2, velocity estimation methods are presented. Section 3 describes the experimental set-up and the data analysed for the performance assessment; in Section 4, the results are discussed. Finally, Section 5 concludes the paper.

## 2. Velocity Estimation by GNSS

In this section, the overall approaches adopted for the evaluation of the velocity by GNSS are introduced.

### 2.1. Doppler-Based Velocity

The most common way to estimate velocity by GNSS is based on the Doppler effect. Specifically, GNSS satellites transmit carrier signals at a nominal frequency, but they are received at a shifted frequency owing to the relative motion between the satellite and receiver. The motion of the satellites is known, being continuously tracked by the ground segment, so the motion of the receiver can be retrieved by measuring the Doppler shift. A detailed description of the Doppler-based algorithm for velocity estimation is reported in [15].

The Doppler shift, D, can be expressed in a unit of velocity by multiplying it by the signal wavelength, λ, as follows:(1)$$\dot{P}=-\lambda \cdot D$$
where $\dot{P}$ is the variation over time of the range between the receiver and satellite.

$\dot{P}$ differs from the geometric range rate between receiver and satellite, $\dot{d}$, because of the clock drifts of the receiver and satellite; moreover, additional error sources include the troposphere, ionosphere, orbits, multipath, and noise. The satellite clock drift is corrected by using information contained in the navigation message, while the other error sources are generally negligible, except for the receiver clock drift, ${\dot{b}}_{u}$, which must be estimated. Therefore, from the Doppler shift the following expression can be obtained:(2)$$\dot{P}=\dot{d}+{\dot{b}}_{u}+\varepsilon_{D}$$
where the term $\varepsilon_{D}$ includes the uncorrected and residual errors.

The geometric range rate $\dot{d}$ can be expressed as
(3)$$\dot{d}=\left({\underline{V}}_{S}-{\underline{V}}_{u}\right)\cdot \hat{e}$$
where ${\underline{V}}_{S}=\left[V_{Sx},V_{Sy},V_{Sz}\right]$ and ${\underline{V}}_{u}=\left[V_{ux},V_{uy},V_{uz}\right]$ are the velocity of the satellite and receiver, respectively, and $\hat{e}=\left[e_{x},e_{y},e_{z}\right]$ is the unit vector of the direction from the receiver to satellite.

The components of $\hat{e}$ can be expressed as $e_{x}=\frac{x_{S}-x_{u}}{d}$, $e_{y}=\frac{y_{S}-y_{u}}{d}$, and $e_{z}=\frac{z_{S}-z_{u}}{d}$, where $\left[x_{S},y_{S},z_{S}\right]$ are the satellite coordinates, $\left[x_{u},y_{u},z_{u}\right]$ are the receiver coordinates and d is geometric distance between the receiver and satellite.

Replacing (3) in (2), rearranging the terms and neglecting the errors in $\varepsilon_{D}$, we obtain
(4)$$\dot{P}-{\underline{V}}_{S}\cdot \hat{e}=-{\underline{V}}_{u}\cdot \hat{e}+{\dot{b}}_{u}$$

Expanding the scalar products, Equation (4) becomes:(5)$$\dot{P}-\left(V_{Sx}\frac{x_{S}-x_{u}}{d}+V_{Sy}\frac{y_{S}-y_{u}}{d}+V_{Sz}\frac{z_{S}-z_{u}}{d}\right)=-\left(V_{ux}\frac{x_{S}-x_{u}}{d}+V_{uy}\frac{y_{S}-y_{u}}{d}+V_{uz}\frac{z_{S}-z_{u}}{d}\right)+{\dot{b}}_{u}$$

For the velocity estimation, the position of the receiver is considered known; similarly, the position and the velocity of the satellite are known as its coordinates are obtained from information in the navigation message (in particular, the details of the computation of ${\underline{V}}_{S}$ are in [24]).

The left-side of (5) contains known terms; the right-side contains the unknowns of the problem, i.e., the coordinates of the user velocity $\left[V_{ux},V_{uy},V_{uz}\right]$ and the receiver clock drift, ${\dot{b}}_{u}$.

The unknowns can be estimated by utilising at least four equations like (5), that is, by measuring the Doppler shift from at least four satellites, simultaneously. In case of redundant measurements, the system of equations can be solved by the least squares method or by another estimation technique. In this study, a Weighted Least Squares method is used, with weight related to the satellite elevation. In a benign environment, Doppler-based velocity is estimated in the order of cm/s. In Figure 1, the scheme of the computation of the Doppler-based velocity is shown.

### 2.2. Carrier Phase-Based Velocity

As an alternative to using the Doppler shift, velocity can be estimated by processing carrier phase measurements differenced in time. The carrier phase equation is shown in (6).
(6)$$\lambda \phi =d+b_{u}-b_{S}+\lambda N+\delta O-\delta I+BGD+\delta T+\delta R+\delta mp+\delta n+\varepsilon_{\phi }$$
where ϕ is the carrier phase measurement in cycles, λ is the wavelength, d is the receiver–satellite geometric distance, $b_{u}$ is the receiver clock bias, $b_{S}$ is the satellite clock bias, N is the integer ambiguity, δO is the orbital error, δI is the ionospheric error, BGD is the Broadcast Group Delay, δT is the tropospheric error, δR is the relativistic error, δmp is the error related to the multipath phenomenon, δn is the error related to the receiver noise, and $\varepsilon_{\phi }$ includes the unmodelled error sources.

Differencing successive carrier phase measurements from the same satellite, integer ambiguity is eliminated and time-correlated error sources, such as $b_{S}$, δO, δI, BGD, δT, and δR, are strongly reduced. This is especially valid when the sampling interval is small, such as 1 Hz or smaller.

The difference between the carrier phase measurements at two consecutive epochs, $T_{0}$ and $T_{1}$, is the TDCP observable, $\lambda \Delta \phi =\lambda \phi \left(T_{1}\right)-\lambda \phi \left(T_{0}\right)$, whose expression is below.
(7)$$\lambda \Delta \phi =\Delta d+\Delta b_{u}+\varepsilon_{\Delta \phi }$$
where Δd and $\Delta b_{u}$ are the differences at consecutive epochs between the geometric distance d and receiver clock bias $b_{u}$, respectively, that is, $\Delta d=d\left(T_{1}\right)-d\left(T_{0}\right)$ and $\Delta b_{u}=b_{u}\left(T_{1}\right)-b_{u}\left(T_{0}\right)$. The terms relative to the multipath and receiver noise, and the residuals of the partially corrected error sources are combined in $\varepsilon_{\Delta \phi }$.

If ${\underline{r}}_{S}$ and ${\underline{r}}_{u}$ are the coordinates of satellite and receiver, respectively, $d\left(T_{0}\right)$ and $d\left(T_{1}\right)$ can be expressed as:(8)$$d\left(T_{0}\right)=\hat{e}\left(T_{0}\right)\cdot \left[{\underline{r}}_{S}\left(T_{0}\right)-{\underline{r}}_{u}\left(T_{0}\right)\right]\;d\left(T_{1}\right)=\hat{e}\left(T_{1}\right)\cdot \left[{\underline{r}}_{S}\left(T_{1}\right)-{\underline{r}}_{u}\left(T_{1}\right)\right]\;$$

The receiver position at the current epoch, ${\underline{r}}_{u}\left(T_{1}\right)$, can be considered as the position at the previous epoch, ${\underline{r}}_{u}\left(T_{0}\right)$, updated considering the displacement ${\underline{\Delta r}}_{u}$, that is, ${\underline{r}}_{u}\left(T_{1}\right)={\underline{r}}_{u}\left(T_{0}\right)+{\underline{\Delta r}}_{u}$; taking this into consideration, the expression of Δd can be manipulated as follows:(9)$$\Delta d=d\left(T_{1}\right)-d\left(T_{0}\right)=\hat{e}\left(T_{1}\right)\cdot \left[{\underline{r}}_{S}\left(T_{1}\right)-{\underline{r}}_{u}\left(T_{1}\right)\right]-\hat{e}\left(T_{0}\right)\cdot \left[{\underline{r}}_{S}\left(T_{0}\right)-{\underline{r}}_{u}\left(T_{0}\right)\right]=\;\left[\hat{e}\left(T_{1}\right)\cdot {\underline{r}}_{S}\left(T_{1}\right)-\hat{e}\left(T_{0}\right)\cdot {\underline{r}}_{S}\left(T_{0}\right)\right]-\left[\hat{e}\left(T_{1}\right)\cdot {\underline{r}}_{u}\left(T_{1}\right)-\hat{e}\left(T_{0}\right)\cdot {\underline{r}}_{u}\left(T_{0}\right)\right]=\;\left[\hat{e}\left(T_{1}\right)\cdot {\underline{r}}_{S}\left(T_{1}\right)-\hat{e}\left(T_{0}\right)\cdot {\underline{r}}_{S}\left(T_{0}\right)\right]-\left[\hat{e}\left(T_{1}\right)\cdot {\underline{r}}_{u}\left(T_{0}\right)+\hat{e}\left(T_{1}\right)\cdot \Delta {\underline{r}}_{u}-\hat{e}\left(T_{0}\right)\cdot {\underline{r}}_{u}\left(T_{0}\right)\right]=\;\left[\hat{e}\left(T_{1}\right)\cdot {\underline{r}}_{S}\left(T_{1}\right)-\hat{e}\left(T_{0}\right)\cdot {\underline{r}}_{S}\left(T_{0}\right)\right]-\left[\hat{e}\left(T_{1}\right)\cdot {\underline{r}}_{u}\left(T_{0}\right)-\hat{e}\left(T_{0}\right)\cdot {\underline{r}}_{u}\left(T_{0}\right)\right]-\hat{e}\left(T_{1}\right)\cdot \Delta {\underline{r}}_{u}\;$$

Setting $\Delta D=\hat{e}\left(T_{1}\right)\cdot {\underline{r}}_{S}\left(T_{1}\right)-\hat{e}\left(T_{0}\right)\cdot {\underline{r}}_{S}\left(T_{0}\right)$ and $\Delta g=\hat{e}\left(T_{1}\right)\cdot {\underline{r}}_{u}\left(T_{0}\right)-\hat{e}\left(T_{0}\right)\cdot {\underline{r}}_{u}\left(T_{0}\right)$, Equation (9) becomes
(10)$$\Delta d=\Delta D-\Delta g-\hat{e}\left(T_{1}\right)\cdot \Delta {\underline{r}}_{u}\;$$
where ΔD is a range variation, which is proportional to the average Doppler frequency shift due to satellite motion along the receiver–satellite direction, and Δg takes into account the changes in receiver–satellite geometry. Replacing expression (10) in (7) and with manipulation, the TDCP measurement equation is obtained:(11)$$\lambda \Delta \phi -\Delta D+\Delta g=-\hat{e}\left(T_{1}\right)\cdot \Delta {\underline{r}}_{u}+\Delta b_{u}+\varepsilon_{\Delta \phi }\;$$

In Equation (11), $\lambda \Delta \phi_{corr}=\lambda \Delta \phi -\Delta D+\Delta g$ is the corrected measurement, while in the right-side of the equation, there are the four unknowns, $\Delta {\underline{r}}_{u}$ and $\Delta b_{u}$. A system of at least four equations like (11) can be solved to estimate the unknowns. Finally, the average velocity between the epochs $T_{0}$ and $T_{1}$, v, is obtained as
(12)$$v=\frac{\Delta {\underline{r}}_{u}}{T_{1}-T_{0}}\;$$

In a benign environment, the TDCP velocity v is estimated at an order of mm/s. The scheme for TDCP velocity estimation is shown in Figure 2.

## 3. Experimental Setup

In order to evaluate the performance of the different methods, data collection was performed on 22 December 2022 in static conditions using a Septentrio Polar Rx5 receiver [25]. The receiver is a high-end receiver able to track GPS, Galileo, GLONASS and Beidou. In particular, the receiver provides the measurements for all the Galileo frequencies (E1, E5a, E5b, E5 Alt BOC and E6). The receiver was connected to geodetic Zephir antenna located on the roof of the European Microwave Signature Laboratory (EMSL) hosting the Joint Research Centre (JRC) testing and demonstration hub for the EU GNSS programmes [26], at the Ispra site of the JRC of the European Commission. About 24 h of data at 1 Hz were collected.

The antenna was located in open-sky conditions, the number of visible satellites and the horizontal dilution of precision (HDOP) time evolution are shown in Figure 3. From this figure, it can be noted that a high number of Galileo satellites (between 6 and 10) were available during the whole data collection period; an average of about 7.82 satellites were visible during the test. In order to consider the geometric conditions during the data collection, the HDOP values are shown in Figure 3 (yellow line with values reported on the right y axis). The HDOP values vary between 1.44 and 2.68 with an average value of about 1.9. The satellite availability and the geometric conditions are typical of open-sky users.

## 4. Results

In this section, the considered configurations and the used metrics are presented. Then, the experimental results are discussed.

### 4.1. Configurations and Metrics

Ten configurations are analysed; they are obtained combining the two velocity estimation methods (Doppler and TDCP) and the five available frequencies (E1, E5a, E5b, E5BOC, and E6):Velocity estimation using Doppler measurements on E1 (Doppler E1);Velocity estimation using carrier phase on E1 (TDCP E1);Velocity estimation using Doppler measurements on E5a (Doppler E5a);Velocity estimation using carrier phase on E5a (TDCP E5a);Velocity estimation using Doppler measurements on E5b (Doppler E5b);Velocity estimation using carrier phase on E5b (TDCP E5b);Velocity estimation using Doppler measurements on E5 Boc (Doppler E5Boc);Velocity estimation using carrier phase on E5 Boc (TDCP E5Boc);Velocity estimation using Doppler measurements on E6 (Doppler E6);Velocity estimation using carrier phase on E6 (TDCP E6).

The performance is assessed in terms of horizontal and vertical velocity errors. The metrics used to assess the performance are mean, standard deviation, and 95th percentile. Finally, the cumulative distribution function (CDF) of the errors is considered.

### 4.2. Experimental Results

The first difference between the Doppler-based velocity estimation and the TDCP velocity estimation is the measurement availability. The TDCP technique requires the availability of two consecutive carrier phase measurements, while the Doppler method only relies on the availability of a single epoch observation. This aspect is analysed in Figure 4, where the number of used measurements for the different methods, considering E1 frequency, is shown. From the figure, it can be noted that the number of measurements for the TDCP case is always lower than that for the Doppler case, which is due to the intrinsic nature of the time difference approach which needs two consecutive measurements. The average number of used measurements for the Doppler case is 7.82, while for the TDCP method, the value is reduced at 7.53. This element, in open-sky conditions, does not significantly impact the performance in terms of solution accuracy; however, in obstructed scenarios where the satellite tracking is not continuous and the carrier phase availability is more limited, it could reduce the solution availability of the TDCP method. In this test, both methods were able to provide a continuous solution during the whole session. To avoid repetition of similar results, the analysis for the other frequencies is not shown.

In Figure 5 and Figure 6, the time evolution of the horizontal velocity error is shown for all the considered frequencies using the Doppler-based approach (blue line) and TDCP approach (red line). In Figure 5, the E1 (upper box) and E6 (lower box) cases are considered, and the three cases related to E5a, E5b and E5 AltBOC are considered in Figure 6 in the upper, central and lower boxes, respectively. The results are very similar for all the cases: the Doppler-based velocity has larger errors with maximum values about 0.5 m/s for the E1, E5a, E5b and E6 cases, and the error is reduced to about 0.2 m/s when AltBOC measurements are used. A substantial reduction in the error can be noted when changing from the Doppler to carrier phase-derived velocity: for the TDCP cases, the maximum horizontal velocity error varies between 0.021 m/s (AltBOC case) and 0.026 m/s (E6 case). For all the considered frequencies, a clear reduction in the horizontal velocity error can be observed. The TDCP method provides improved performance: the red line is always lower than the blue line.

The time evolution of the vertical velocity error is shown in Figure 7 and Figure 8 for all the considered frequencies. In Figure 7, the E1 (upper box) and E6 (lower box) cases are considered; in Figure 8, the three cases related to E5 frequencies are shown: upper part E5a, central box E5b, and lower box E5AltBOC. The results are very similar to the horizontal channel, and also in this case, the Doppler-based velocity has a larger error. Also, in the vertical channel, the most accurate solution for the Doppler measurement cases is obtained using E5AltBOC observables; indeed, a maximum error of about 30 cm/s was observed, while a larger error about 0.7 m/s was obtained with the other frequencies. When the TDCP approach is used, the maximum vertical velocity error varies between 0.013 m/s (AltBOC case) and 0.031 m/s (E1 case). For all the considered frequencies, a visible reduction in the vertical velocity error can be noted.

In Figure 9 and Figure 10, the statistical parameters used to summarize the performance of the two approaches are shown, for the horizontal and vertical errors, respectively. In the upper box the mean values are reported; in the central box the standard deviation values are shown; finally, in the lower box the 95th percentile is considered. From the upper plot of Figure 9, it can be noted that for the Doppler-based velocity, the mean error is reduced using E5AltBOC measurements and the largest values are obtained using E1 and E6 measurements. A similar trend can be observed when considering the standard deviation (central plot) and the 95th percentile. In particular, the advantages of using E5AltBOC Doppler observables clearly emerges in the standard deviation case: the solution obtained using AltBOC observables is the only one with a standard deviation lower than 10 cm/s. Comparing blue and red bars, the advantages of the TDCP method clearly emerge for all the considered parameters. For the TDCP case, no specific improvements can be noted using AltBOC measurements; indeed, only differences in the order of submillimetres per second are observed among the solutions: the mean errors are within the interval 2.4 mm/s (AltBOC case) and 2.7 mm/s (E6 case). In terms of standard deviations, all the solutions have a very similar value of about 1 mm/s. Finally, in the 95th percentile case, the values for the TDCP methods are about 5 mm/s, while for the Doppler case, the values are between 32 cm/s (AltBOC case) and 57 cm/s (E1 case), confirming the large variability in the Doppler-based velocity estimation.

In Figure 10, the statistical parameters of the vertical velocity error are shown. Comparing the dark orange bars, it can be noted that the AltBOC solution is the one with the lowest values in terms of mean, standard deviation and 95th percentile, while the solution obtained using E1 measurements has the largest values. In terms of mean and standard deviation, the AltBOC solution is the only solution with values lower than 20 cm/s (16 cm/s and 14 cm/s, respectively). For the TDCP method, considerations similar to the horizontal case can be drawn: no specific advantages of using the different measurements can be noted, and a clear reduction in all the parameters (with respect to the Doppler-based solution) can be observed.

In Figure 11 and Figure 12, the respective CDF of the horizontal and vertical velocity errors are shown. In both figures, the left boxes show the CDF of the Doppler-based solutions, while, in the right boxes, the TDCP solutions are considered. The frequency cases are represented with different line colours. For the vertical velocity errors, the absolute value is considered for this plot. From the CDF of the Doppler-based solutions (left boxes) it can be noted that the E1 and E6 curves are almost superimposed, and a reduction in the error is obtained when changing to E5 frequency. In this case, E5a and E5b have very similar performance: the yellow and purple lines are very close. Finally, the highest curve is the one related to E5 AltBOC, confirming the results observed in the previous graphs. For the TDCP case, all the curves are almost overlapping and only very small differences can be observed. The CDF plots also confirm the benefits of using the TDCP method with respect to the Doppler-based solution.

## 5. Conclusions

Velocity estimation could play a key role in fulfilling the European Commission’s regulations regarding eCall and smart tachographs. For this reason, in this work, the velocity estimation performance of the European GNSS, Galileo, are explored. In particular, two methods are considered: one based on Doppler observables, and one based on time differences in the carrier phases, i.e., TDCP. All the signals available from Galileo are taken into account, that is, E1, E6, E5a, E5b and E5AltBOC. The performance of the two methods is assessed considering horizontal and vertical velocity errors; the statistical parameters used are mean, standard deviation and 95th percentile.

A static test of 24 h was carried out in an open-sky scenario with geodetic equipment. In accordance with the theory, TDCP-based velocity resulted in errors in the order of a few mm/s. No specific advantages could be noted by using any of the different signals. For the horizontal channel, the mean error was within the interval between 2.4 mm/s (AltBOC case) and 2.7 mm/s (E6 case). The standard deviations of all the TDCP solutions have a very similar value of about 1 mm/s.

Doppler-based velocity demonstrated errors one order of magnitude larger, that is, of a few cm/s. In this case, the different signals provided significantly different performance. Specifically, significant benefits are evident with E5AltBOC measurements, which outperform E5a and E5b measurements, and still more E1 and E6 measurements. In terms of mean and standard deviation for the horizontal channel, the AltBOC solution was the only solution with values lower than 20 cm/s (16 cm/s and 14 cm/s, respectively).

From the analysis, it emerged that for a more accurate velocity solution, the TDCP methods guarantees a lower error. If the user does not have reliable and continuous carrier phases, the Doppler solution with the lower errors is the one based on the E5AltBOC Doppler shift.

In this work, only a static test was carried out due to the unavailability of a reference solution with sub-mm/s accuracy, which would be necessary for assessing the TDCP velocity performance. A kinematic test would be more stressful for the GNSS velocity estimation, especially for the TDCP technique; indeed, in such situation, the vehicle dynamics and the changing scenario (from open-sky to urban and vice versa) would introduce multipath-related blunders and cycle slips. For these reasons, the future development of this research will surely include kinematic tests.

## Figures and Tables

**Figure 1 sensors-23-06828-f001:**
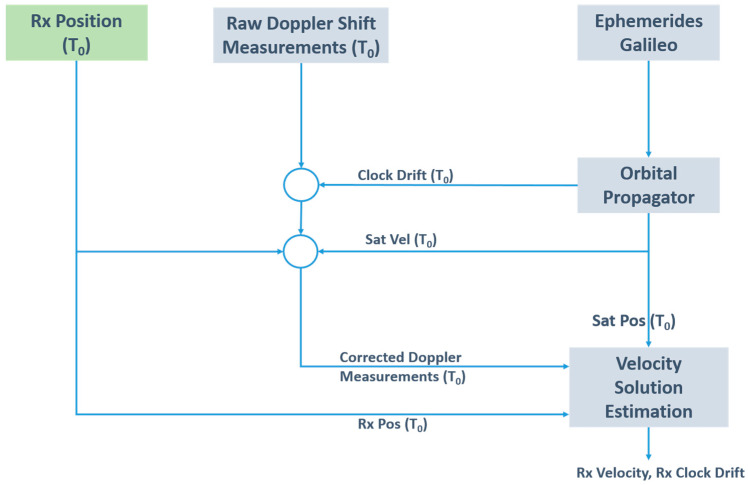
Doppler-based velocity diagram.

**Figure 2 sensors-23-06828-f002:**
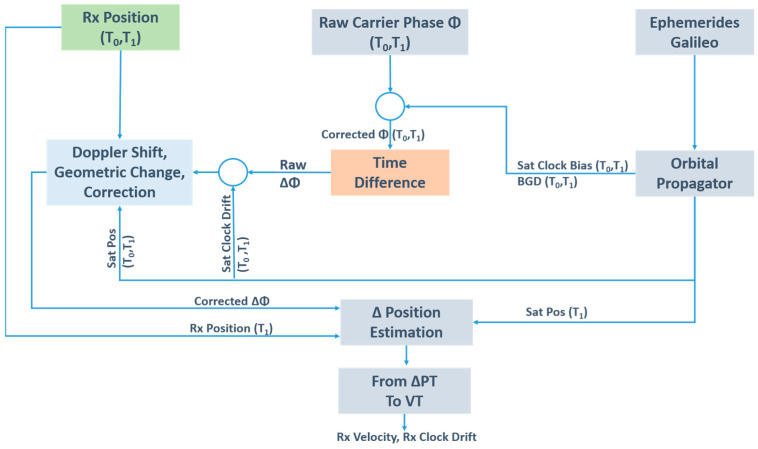
TDCP velocity diagram.

**Figure 3 sensors-23-06828-f003:**
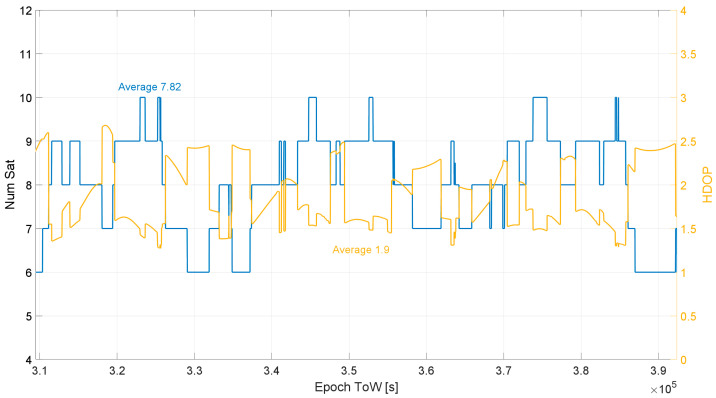
Number of visible satellites and HDOP values observed during the test on 22 December 2022.

**Figure 4 sensors-23-06828-f004:**
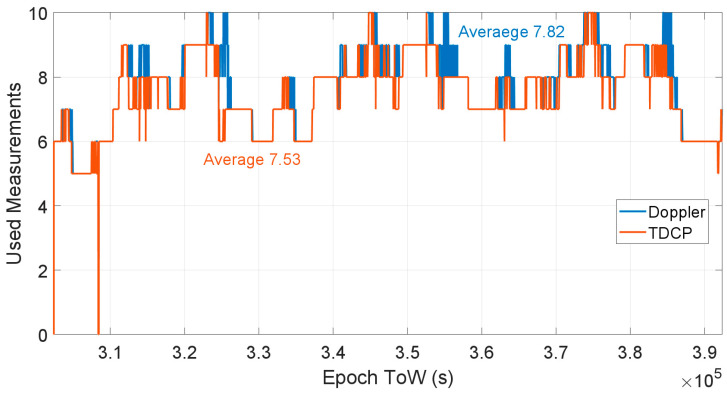
Number of used measurements for Doppler and TDCP methods considering E1 frequency.

**Figure 5 sensors-23-06828-f005:**
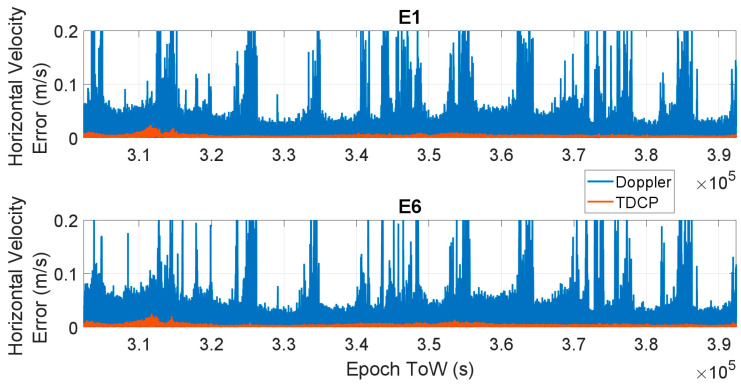
Horizontal Velocity errors as a function of time considering Doppler-based (blue line) and TDCP (red line) methods. Upper box: E1 frequency, lower box: E6 frequency.

**Figure 6 sensors-23-06828-f006:**
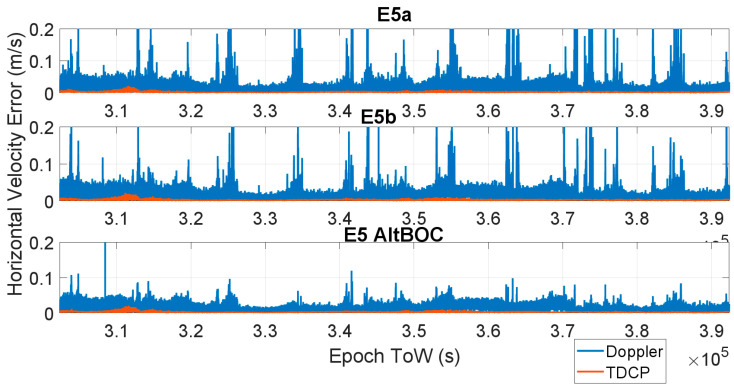
Horizontal velocity errors as a function of time considering Doppler-based (blue line) and TDCP (red line) methods. Upper box: E5a frequency, central box: E5b frequency, and lower box: E5 AltBOC frequency.

**Figure 7 sensors-23-06828-f007:**
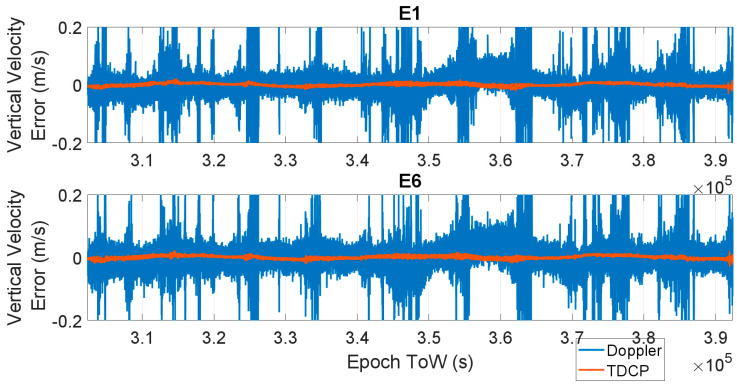
Time evolution of the vertical velocity error considering Doppler-based (blue line) and TDCP (red line) methods. Upper box: E1, lower box: E6.

**Figure 8 sensors-23-06828-f008:**
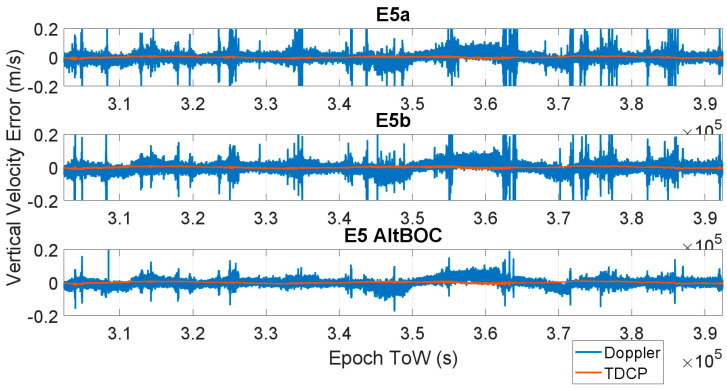
Vertical velocity errors as a function of time for the three E5 cases: upper box, E5a; central box, E5b; and lower box, E5 AltBOC. Doppler-based (blue line) and TDCP (red line).

**Figure 9 sensors-23-06828-f009:**
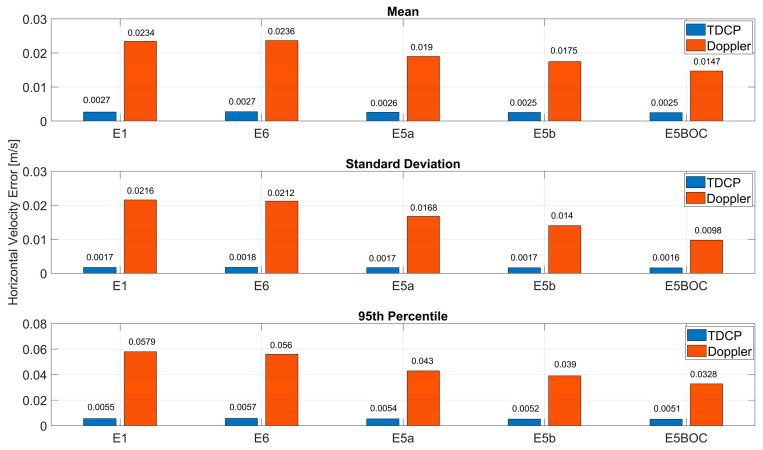
Statistical parameters of the horizontal velocity errors for all the frequencies considering Doppler-based and TDCP solutions.

**Figure 10 sensors-23-06828-f010:**
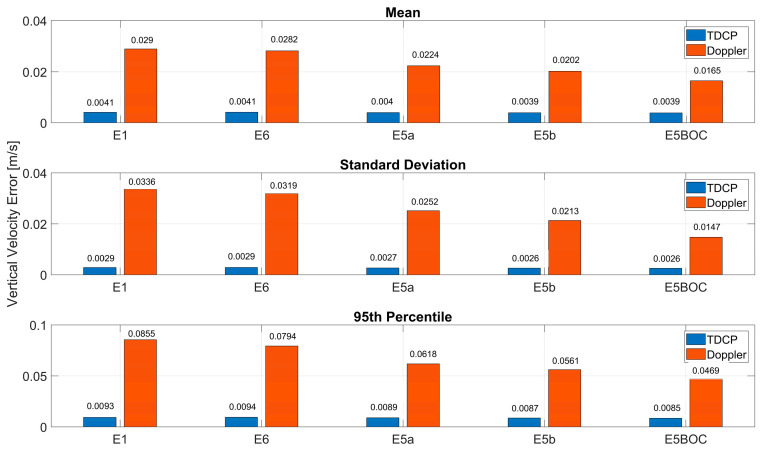
Statistical parameters of the vertical velocity errors for all the frequencies considering Doppler-based and TDCP solutions.

**Figure 11 sensors-23-06828-f011:**
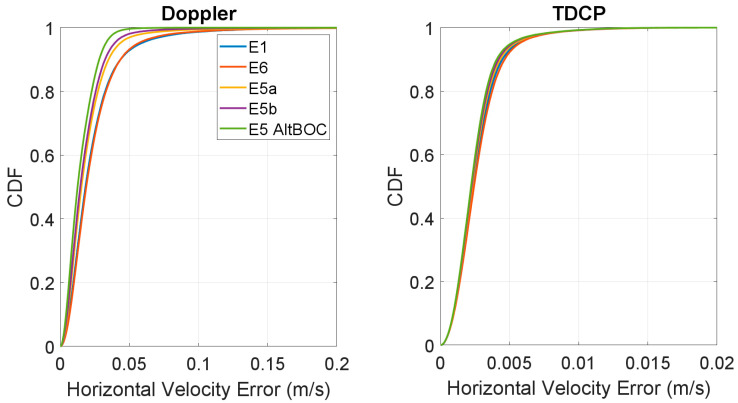
CDF of the horizontal velocity error for the Doppler-based (**left box**) and TDCP (**right box**) methods.

**Figure 12 sensors-23-06828-f012:**
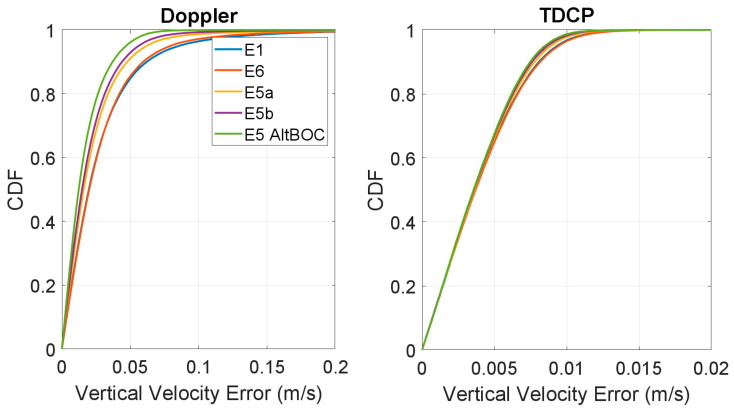
CDF of the absolute values of the vertical velocity error for the Doppler-based (**left box**) and TDCP (**right box**) methods.

## Data Availability

The data used in the study will be provided by the authors available on request.

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
