# Peer review of "Galileo-Based Doppler Shifts and Time Difference Carrier Phase: A Static Case Demonstration"

_sensors, 2023, doi:10.3390/s23156828_

Round 1

Reviewer 1 Report

Global Navigation Satellite System (GNSS) to compute position velocity and time (PVT) information has been exploited for many years. The safety of road users is increased with Intelligent Speed Assistance (ISA) devices based on GNSS. With this background, the velocity of the vehicle plays a fundamental rule. Two novel and interesting methods for velocity estimation using Galileo are presented and compared in the paper. It firstly exploits Doppler shift measurements, and the other uses time difference carrier phase (TDCP) measurements; the methods are demonstrated using all Galileo frequencies. Regarding the results, the methods are compared in terms of velocity errors for both horizontal and vertical components using real data. Overall, the paper is novel and results are solid to support the contributions of the paper. Therefore, it is a good paper but still requires some minor revisions. Below are my comments.

1) In the introduction, the GPS is also an important source for time synchronization based on the PPS when fusing with LiDAR and camera for vehicle/robot navigation. Some related work should be included to compare the pros and cons between camera and LiDAR: an automated driving systems data acquisition and analytics platform, hydro-3d: hybrid object detection and tracking for cooperative perception using 3d lidar, yolov5-tassel: detecting tassels in rgb uav imagery with improved yolov5 based on transfer learning. With deep analysis, it would help to highlight the contribution of your work.

2) GNSS is pretty important for vehicle safety, as it could provide the vehicle position, velocity, and attitude with the fusion of IMU and other sensors’ signal. Some related work solves this issue such as: autonomous vehicle kinematics and dynamics synthesis for sideslip angle estimation based on consensus kalman filter, vehicle sideslip angle estimation by fusing inertial measurement unit and global navigation satellite system with heading alignment, imu-based automated vehicle body sideslip angle and attitude estimation aided by gnss using parallel adaptive kalman filters, automated vehicle sideslip angle estimation considering signal measurement characteristic. The above references should be included in the introduction to expand the application of GPS.

3) Please provide the work limitation and future work at the end.

4) At the end of the introduction, please list your contributions to this work.

5) Please optimize Figure 3, there is a dotted line at the bottom of the figure.

Author Response

General Comment:

Global Navigation Satellite System (GNSS) to compute position velocity and time (PVT) information has been exploited for many years. The safety of road users is increased with Intelligent Speed Assistance (ISA) devices based on GNSS. With this background, the velocity of the vehicle plays a fundamental rule. Two novel and interesting methods for velocity estimation using Galileo are presented and compared in the paper. It firstly exploits Doppler shift measurements, and the other uses time difference carrier phase (TDCP) measurements; the methods are demonstrated using all Galileo frequencies. Regarding the results, the methods are compared in terms of velocity errors for both horizontal and vertical components using real data. Overall, the paper is novel and results are solid to support the contributions of the paper. Therefore, it is a good paper but still requires some minor revisions. Below are my comments.

Comment 1:
In the introduction, the GPS is also an important source for time synchronization based on the PPS when fusing with LiDAR and camera for vehicle/robot navigation. Some related work should be included to compare the pros and cons between camera and LiDAR: an automated driving systems data acquisition and analytics platform, hydro-3d: hybrid object detection and tracking for cooperative perception using 3d lidar, yolov5-tassel: detecting tassels in rgb uav imagery with improved yolov5 based on transfer learning. With deep analysis, it would help to highlight the contribution of your work.

Authors Answer:

Thank you for point out the synchronization issue. The authors mentioned such aspect in the introduction. To further support the aspect additional references have been included.

In the introduction, a small section has been added to include non GNSS-based approach, in particular pros and cons of LiDAR and camera for vehicle and robot are considered. The following references has been added to the manuscript:

  • Xin Xia, Zonglin Meng, Xu Han, Hanzhao Li, Takahiro Tsukiji, Runsheng Xu, Zhaoliang Zheng, Jiaqi Ma, An automated driving systems data acquisition and analytics platform, Transportation Research Part C: Emerging Technologies, Volume 151, 2023, 104120, ISSN 0968-090X, https://doi.org/10.1016/j.trc.2023.104120.
  • Meng, X. Xia, R. Xu, W. Liu and J. Ma, "HYDRO-3D: Hybrid Object Detection and Tracking for Cooperative Perception Using 3D LiDAR," in IEEE Transactions on Intelligent Vehicles, doi: 10.1109/TIV.2023.3282567.
  • Liu, K. Quijano and M. M. Crawford, "YOLOv5-Tassel: Detecting Tassels in RGB UAV Imagery With Improved YOLOv5 Based on Transfer Learning," in IEEE Journal of Selected Topics in Applied Earth Observations and Remote Sensing, vol. 15, pp. 8085-8094, 2022, doi: 10.1109/JSTARS.2022.3206399.
  • Gioia e D. Borio, «Multi-Layer Defences for Robust GNSS Timing Retrieval» Sensors, vol. 23, 2021.
  • Gioia, «T-RAIM Approaches: Testing with Galileo Measurements» Sensors, 2023.

Comment 2:
GNSS is pretty important for vehicle safety, as it could provide the vehicle position, velocity, and attitude with the fusion of IMU and other sensors’ signal. Some related work solves this issue such as: autonomous vehicle kinematics and dynamics synthesis for sideslip angle estimation based on consensus kalman filter, vehicle sideslip angle estimation by fusing inertial measurement unit and global navigation satellite system with heading alignment, imu-ased automated vehicle body sideslip angle and attitude estimation aided by gnss using parallel adaptive kalman filters, automated vehicle sideslip angle estimation considering signal measurement characteristic. The above references should be included in the introduction to expand the application of GPS.

Author answer:

The authors thank the reviewer for highlighting the documents and their relevance for the topic. The following references have been added to the manuscript:

  • Xia, E. Hashemi, L. Xiong and A. Khajepour, "Autonomous Vehicle Kinematics and Dynamics Synthesis for Sideslip Angle Estimation Based on Consensus Kalman Filter," in IEEE Transactions on Control Systems Technology, vol. 31, no. 1, pp. 179-192, Jan. 2023, doi: 10.1109/TCST.2022.3174511.
  • Xin Xia, Lu Xiong, Yishi Lu, Letian Gao, Zhuoping Yu, Vehicle sideslip angle estimation by fusing inertial measurement unit and global navigation satellite system with heading alignment, Mechanical Systems and Signal Processing, Volume 150, 2021, 107290, ISSN 0888-3270, https://doi.org/10.1016/j.ymssp.2020.107290.
  • Liu, X. Xia, L. Xiong, Y. Lu, L. Gao and Z. Yu, "Automated Vehicle Sideslip Angle Estimation Considering Signal Measurement Characteristic," in IEEE Sensors Journal, vol. 21, no. 19, pp. 21675-21687, 1 Oct.1, 2021, doi: 10.1109/JSEN.2021.3059050.

Comment 3:
Please provide the work limitation and future work at the end.

Author answer:

The work limitation and the future work have been included in the Conclusions section.

Comment 4:
At the end of the introduction, please list your contributions to this work.

Author answer:

The main contributions of the work have been better presented in the introduction.

Comment 5:
Please optimize Figure 3, there is a dotted line at the bottom of the figure.

Author answer:

Figure 3 has been replaced, currently the authors do not see any issue with the figure. If the reviewer is looking only at the PDF file the issue could be due to the word-pdf conversion.

Detailed replies to the editors/reviewers’ comments are provided in the attached file.

Reviewer 2 Report

The manuscript is well written and structured. Author provide all necessary information for understanding method they use to obtain results. Introduction contains motivation for current study: “In the event of an accident, in-vehicle sensors will automatically trigger an emergency call”. So the issue is clear: accurate velocity estimation are needed to improve the accuracy of “trigger” to reduce false-negative and false-positive scenarios. However in my opinion inertial measurements units (IMU) should perform this task better, and they are not affected by GNSS availability (satellite visibility). I recommend to include the description of any “in-vehicle sensors” system used available at market to appreciate the importance of accurate velocity estimation from GNSS measurements.

The manuscript entitled “Galileo-based velocity estimation methods...” however paper contains only static experiment (or I didn’t find its description). So the methods of velocity estimation are applied only to the cases when velocity is zero. Static conditions reduce many issues that we encounter in real world, such as cycle slips (that author mentioned in introduction and didn’t considered further in the manuscript) or multi-path propagation. These effects could be significant to the moving receiver making current method not applicable to dynamic experiment where we really need velocity estimation.

The manuscript needs additional experiments to fit the goal author stated.

Line 20: “fundamental rule” → “fundamental role”

Is it a typo?

Author Response

General Comment:

The manuscript is well written and structured. Author provide all necessary information for understanding method they use to obtain results. Introduction contains motivation for current study: “In the event of an accident, in-vehicle sensors will automatically trigger an emergency call”. So the issue is clear: accurate velocity estimation are needed to improve the accuracy of “trigger” to reduce false-negative and false-positive scenarios.

However in my opinion inertial measurements units (IMU) should perform this task better, and they are not affected by GNSS availability (satellite visibility). I recommend to include the description of any “in-vehicle sensors” system used available at market to appreciate the importance of accurate velocity estimation from GNSS measurements.

Author answer

The possible use of other sensors, like inertial sensor, is mentioned and supported by bibliographic references. See author answer to Reviewer 1 Comment 2 that includes the list of references added in the current version of the paper.

Comment 1:

The manuscript entitled “Galileo-based velocity estimation methods...” however paper contains only static experiment (or I didn’t find its description). So the methods of velocity estimation are applied only to the cases when velocity is zero. Static conditions reduce many issues that we encounter in real world, such as cycle slips (that author mentioned in introduction and didn’t considered further in the manuscript) or multi-path propagation. These effects could be significant to the moving receiver making current method not applicable to dynamic experiment where we really need velocity estimation. The manuscript needs additional experiments to fit the goal author stated.

Author answer

The authors completely agree with the reviewer; the algorithms (in particular TDCP) would be stressed in a kinematic test and the results would be surely interesting. To properly analyze the TDCP performance in kinematic mode, it would be necessary to have a reference solution with sub-mm/s accuracy and unfortunately, at the moment, the authors don’t have available a suitable equipment. For this reason, in order to have an accurate ground truth (a velocity equal to zero), a static test was performed.

In the introduction, the following part has been added to clarify the testing conditions:

‘The performance of the methods is assessed using real data collected in static conditions. A long data collection of about 24 hours using a professional receiver has been used for testing. Although, static conditions reduce the presence of issues such as cycle slips and multipath propagation they have the advantages of an accurate reference solution allowing a proper error characterization for the methods presented.’

In addition, the limit of the work was remarked in the conclusions section and the analysis of a kinematic test was postponed for a future work.

‘In this work, only a static test was carried out due to the unavailability of a reference solution with sub-mm/s accuracy, necessary for assessing the TDCP velocity performance. A kinematic test would be more stressful for the GNSS velocity estimation, especially for TDCP technique; indeed, in such situation, the vehicle dynamics and the changing scenario (from open-sky to urban and vice versa) would introduce multi-path-related blunders and cycle slips. For these reasons, the future development of this research will surely include kinematic tests.’

Comment 2:
Line 20: “fundamental rule” → “fundamental role” Is it a typo?

Author answer

Yes, it was a typo. Now it has been corrected.

Reviewer 3 Report

The author presents and compares two methods for velocity estimation using Galileo including exploits Doppler shift measurements annd time difference carrier phase. Overall, thus work is well organized and presented, but there are some major concerns that need to be fixed:

1. Abstract needs to be enhanced: Too many sentences in the background introduction and the contributions of this work are not detailed.

2. The limitations of current work are required to be discussed in the introduction part, and then discuss main contributions.

3. Velocity estimation by GNSS part, which are the contributions and innovations of this work compared with existing methods? discussion is required.

4. more experimental comparison is needed in this work to verify the effectiveness of the proposed methods.

5. It is recommended to add Limitations and future work discussion in the Conclusion part.

n/a

Author Response

General Comment:

The author presents and compares two methods for velocity estimation using Galileo including exploits Doppler shift measurements and time difference carrier phase. Overall, thus work is well organized and presented, but there are some major concerns that need to be fixed:

Comment 1:
Abstract needs to be enhanced: Too many sentences in the background introduction and the contributions of this work are not detailed.

Author answer

The abstract has been modified according to reviewer suggestions.

Comment 2:
The limitations of current work are required to be discussed in the introduction part, and then discuss main contributions.

Author answer

The paper has been adjusted according to this suggestion. In details, according to the authors, the main limit of this work is the lack of a kinematic test. This limit has been remarked in the “conclusions” section.

Comment 3:
Velocity estimation by GNSS part, which are the contributions and innovations of this work compared with existing methods? discussion is required.

Author answer

The main contribution of the work is, according to the authors, the analysis of the performance of Doppler-based and TDCP velocities with all the Galileo signals (“The authors do not know of any other research which analyses the aforementioned techniques with all the available Galileo signals”). Doppler-based velocity is pretty common, but the existing researches do not analyzed it for all the Galileo signals; a remarkable result is about the E5 Alt BOC performance, which are enhanced with respect to the other signals. On the other side, TDCP velocity is emerging, and several studies have been published about it in the last years; anyway, none of them is about all the Galileo signals.

Comment 4:
more experimental comparison is needed in this work to verify the effectiveness of the proposed methods.

Author answer

Additional tests in static and kinematic mode will be included in future developments of the work. See also author answer to Reviewer 2 Comment 1.

Comment 5:
It is recommended to add Limitations and future work discussion in the Conclusion part.

Author answer

The work limitations and the future work have been included in the Conclusions section.

Round 2

Reviewer 2 Report

I suggest change title and revise abstract so it is include the information that no actual velocity estimation is performed in the manuscript since the only static case is considered. Current manuscript may confuse a reader, who is looking for actual velocity estimation i.e. for moving receiver.

Author Response

The authors wish to thank the editors and the reviewers for the thorough and useful evaluation of their paper. The authors have benefited from the reviewer/editor comments and insights and have revised the paper according to their suggestions.

Detailed replies to the editors/reviewers’ comments are provided in the following.

Reviewer's Comments to Author:
-------------------------------------------

Reviewer 2

Comment 1:
I suggest change title and revise abstract so it is include the information that no actual velocity estimation is performed in the manuscript since the only static case is considered. Current manuscript may confuse a reader, who is looking for actual velocity estimation i.e. for moving receiver.

Authors Answer:

Although the methods presented could be exploited for actual velocity estimation the authors decided to change the title of the manuscript to better represent the content of the paper. The new proposed title is:

Galileo-based Doppler shifts and time difference carrier phase: a static case demonstration.

The abstract has been modified including a clear indication on the static data used for the methods demonstration.

Reviewer 3 Report

The author has addressed all my concerns, hence, I am glad to recommed this paper to be accepted at its current form.

n/a

Author Response

The authors wish to thank the editors and the reviewers for the thorough and useful evaluation of their paper. The authors have benefited from the reviewer/editor comments and insights and have revised the paper according to their suggestions.